# Fecal Microbiota Transplantation in Animals: Therapeutics, Conservation, and Farming

**DOI:** 10.3390/microorganisms13112465

**Published:** 2025-10-29

**Authors:** Kelly R. Reveles, Joni Meehan, Glenn Tillotson

**Affiliations:** 1College of Pharmacy, The University of Texas at Austin, Austin, TX 78712, USA; revelesk@uthscsa.edu; 2School of Medicine, University of Texas Health San Antonio, San Antonio, TX 78229, USA; 3GST Micro, LLC, North, VA 23128, USA; jonit77@gmail.com

**Keywords:** fecal microbiota transplantation, microbiome, veterinary medicine, animal health, conservation biology

## Abstract

Fecal microbiota transplantation (FMT) is increasingly used in both human and veterinary settings to restore gut microbiota and promote health. Advances in sequencing technologies and microbiome analysis have expanded our understanding of microbial communities and enabled broader FMT applications. As insights grow into what constitutes a healthy microbiome, interest in using FMT across a range of animal contexts has also increased. This narrative review highlights recent progress in the use of FMT to improve the welfare of farm animals, manage infectious and chronic conditions in companion animals, and support the health of wildlife in conservation and reintroduction programs. Representative examples from each domain are discussed.

## 1. Historical Use of Fecal Microbiota Transplantation

The practice of fecal microbiota transplantation (FMT) predates the discovery of microbes and the concept of the gut microbiota. The earliest known records date back to 4th century China, where a fecal preparation known as “yellow soup” was administered to patients with severe diarrhea and food poisoning [1]. By the 16th century, feces-derived treatments had become a common remedy for gastrointestinal conditions such as constipation, diarrhea, and abdominal pain, as well as systemic symptoms like fever. These methods were documented by the Chinese physician Li Shizhen (1518–1593) and included the use of fermented fecal solutions, fresh suspensions, dried feces, and infant feces. This practice—referred to as transfaunation—involves transferring fecal material from a healthy donor to a recipient to help restore gut microbial homeostasis [2].

In Europe, the medicinal use of feces was first suggested by German physician Christian Paullini in 1686. The idea that gut microbes may play a beneficial role in health was further advanced in the late 19th century by Élie Metchnikoff, a Ukrainian bacteriologist known for his pioneering work in immunology and probiotics. The practical significance of gut bacteria was reinforced during World War II, when Nazi medical officers observed that North African nomads who consumed fresh, warm camel dung experienced fewer cases of dysentery. Analysis revealed the presence of *Bacillus subtilis*, a bacterium capable of suppressing pathogenic microbes. Rather than using camel dung directly, the medical corps cultured *B. subtilis* and administered it in broth or powdered form to prevent dysentery among troops [3].

The first formal scientific report of FMT in humans was published in 1958 by Eiseman et al., who successfully treated pseudomembranous colitis using donor stool [4]. Since then, FMT has gained prominence—particularly for the treatment of recurrent *Clostridioides difficile* infection (rCDI)—with success rates of up to 95% reported in early observational studies and clinical trials [5,6,7,8,9,10,11]. The therapy works by restoring healthy microbial communities to the damaged gut [5,6]. Recently, two donor-derived microbiota products have received FDA approval: fecal microbiota, live-jslm (REBYOTA^®^) [12] and fecal microbiota spores, live-brpk (VOWST^®^) [13]. These preparations undergo rigorous screening, including testing for 29 pathogens. Beyond rCDI, FMT is being explored for a wide range of conditions, including inflammatory bowel disease, obesity, metabolic disorders, and neuropsychiatric conditions [14,15]. Early evidence suggests FMT may help restore the gut–brain axis in certain neurological disorders [16].

Advancements in microbiome research over the past decade have dramatically expanded our understanding of microbial ecosystems. Numerous bacterial, viral, fungal, and parasitic species—many of which are not cultivable—have now been identified using metagenomics. Additional “multi-omic” approaches provide further insight into microbial function: metabolomics profiles the chemical products of gut metabolism; metaproteomics identifies and quantifies proteins from complex microbial communities; and metatranscriptomics reveals gene expression patterns of microbes within their natural environments [17]. Together, these tools offer a comprehensive view of microbial structure and function. While FMT has long been used in veterinary and agricultural contexts, these emerging technologies will deepen our understanding of its mechanisms and applications across diverse animal species.

## 2. Veterinary Use of FMT

As our understanding of the gut microbiome has improved, the veterinary field has increasingly adopted FMT as a tool to support animal health and productivity. This section presents a narrative overview of relevant examples drawn from both historical and contemporary literature. While not identified through a formal systematic search, studies were selected through targeted PubMed queries and supplemented by key references cited in recent reviews.

Historically, FMT in veterinary medicine dates back centuries and was first applied in ruminants to treat conditions such as ruminal indigestion. Fabricius Acquapendente (1537–1619), an Italian anatomist and surgeon, is credited with one of the earliest documented uses of gastrointestinal content transfer between animals to restore health. The practice, later referred to as transfaunation, became a widespread treatment for promoting normal rumination in cattle [18]. Roughly 150 years later, Swedish veterinarians Brag and Hansen further advanced the concept by using regurgitated digesta (cud) as a microbial transplant. They highlighted the health-promoting role of “living creatures” within the cud—effectively recognizing the microbial nature of its benefits [18].

Modern veterinary applications of FMT have since expanded beyond large ruminants to include small companion animals, particularly for gastrointestinal conditions such as chronic diarrhea and inflammatory bowel disease (IBD) [3]. Notably, some of the earliest recorded uses of FMT in Europe occurred within veterinary contexts. Here, we discuss the importance of the gastrointestinal tract and its manipulation in the spheres of pets, conservation, and food production. To better understand how FMT is conducted in practice, an overview of preparation and administration steps is shown in Figure 1. The source of the preparation may be influenced by the initial source of the feces. This diagram shows a general approach it may vary by animal being managed. Especially considering the companion animal compared with feed or conservancy animals. It is important to appreciate that not all FMTs are prepared in a similar way, and the various preparations can vary widely across species.

### 2.1. Inflammatory Bowel Disease in Companion Animals

Chronic enteropathies are well recognized in both human and veterinary medicine. In dogs, inflammatory bowel disease (IBD) is diagnosed based on histological evidence of characteristic intestinal inflammation, following exclusion of other causes such as systemic, endocrine, neoplastic, or infectious disease [20]. The interplay between genetic predisposition, environmental factors, immune response, and gut microbiota makes this condition complex and multifactorial.

Canine IBD shares many pathophysiological and phenotypic features with its human counterpart. In fact, the genetic homogeneity within some dog breeds—resulting from inbreeding—resembles isolated human populations and offers a unique opportunity to study complex disorders like IBD. For example, studies in Yorkshire Terriers suggest a breed-specific enteropathy that differs from presentations in other dogs. For example, in a study by Doulidis et al. [21], 16S rRNA sequencing was used to compare the gut microbiota of healthy Yorkshire Terriers to those with IBD. The analysis revealed a distinct dysbiosis pattern, with both overrepresentation and underrepresentation of specific bacterial genera. Notably, even dogs in clinical remission exhibited substantial microbial disruption, suggesting that the gut microbiota may not fully normalize despite symptom resolution. These findings support the potential utility of FMT to restore microbial balance in canine IBD.

A pilot study explored the use of a freeze-dried fecal microbiota supplement—Gut Restore Supplement (GRS)—in 40 dogs and 27 cats with suspected or diagnosed IBD. The supplement was administered as an oral capsule twice daily for 25 days. Clinical improvement was reported in 80% of dogs and 83% of cats, including reduced frequency of diarrhea and improved stool quality. Additionally, increases in appetite were reported in approximately 50% of dogs and 25% of cats, with associated weight gain in some animals [22].

AnimalBiome, the producer of GRS, has published data on their stabilization method for maintaining viable bacterial content in freeze-dried feces. Their process retains high bacterial cell viability—93.5% for canine feces and 90% for feline feces—with concentrations of 20 × 10^9^ and 19 × 10^9^ live bacterial cells per gram, respectively [22]. For comparison, Weese and Martin [23] assessed 25 commercial veterinary probiotics and found actual bacterial counts ranging from zero to 2 × 10^9^ CFU/g. Notably, 1 g of freeze-dried canine feces in GRS contains approximately 20 billion viable cells, highlighting its high microbial load relative to standard probiotics [23].

### 2.2. Clostridioides difficile Infection (CDI)

Several case reports have described the use of FMT in dogs with gastrointestinal disease, including CDI. In one such report, Sugita et al. [24] documented the treatment of a male French Bulldog with a four-month history of intermittent large bowel diarrhea. Diagnostic testing, including polymerase chain reaction testing and immunochromatography, confirmed the presence of *C. difficile* and its associated toxins. The dog received an oral FMT using a fecal suspension from a healthy beagle. Within 2–3 days of treatment, stool consistency and frequency normalized. Follow-up testing was negative for *C. difficile* toxins A and B, and no adverse events were observed. The authors concluded that oral FMT was a safe and effective treatment for CDI in this case and recommended further investigation into its broader use in veterinary medicine [24].

### 2.3. Canine Parvovirus Infection

Canine parvovirus is a highly contagious and potentially fatal disease, commonly associated with severe hemorrhagic diarrhea and high mortality rates in young dogs. In a randomized clinical trial, Pereira et al. [25] evaluated the efficacy of FMT as an adjunct to standard therapy in 66 puppies under one year of age diagnosed with parvovirus infection. The puppies were randomized into two treatment groups: one received standard therapy alone (*n* = 33), and the other received standard therapy plus oral FMT (*n* = 33). Mortality was lower in the FMT group (21.6%) compared to the standard therapy group (36.4%), although the difference did not reach statistical significance (*p* = 0.174). Among surviving puppies, symptom resolution within 48 h was observed in 61.5% of the FMT group compared to only 4.8% in the standard treatment group.

The authors emphasized that the rapid improvement seen with FMT could be clinically meaningful, particularly in resource-limited settings where cost constraints influence treatment decisions. Faster recovery may reduce hospitalization time and the likelihood of euthanasia for economic reasons [25].

### 2.4. Acute Hemorrhagic Diarrheal Syndrome (AHDS)

AHDS is a clinical condition in dogs characterized by sudden onset of vomiting and profuse, bloody diarrhea, often accompanied by abnormal serum protein levels [26]. Although the precise cause remains unknown, AHDS has been associated with bacterial toxins, dietary components, and disruptions in the gut microbiome [27,28,29]. While a direct causal relationship with a specific bacterium or toxin has not been established, the presence of gut dysbiosis in AHDS cases suggests that microbiome-based therapies such as FMT may be beneficial.

In a randomized, placebo-controlled, open-label pilot study, Gal et al. [30] evaluated FMT in dogs with AHDS. The study enrolled eight affected dogs (seven study dogs and one client-owned pet), along with three healthy donor dogs. Donors were screened using criteria aligned with recommendations from the Companion Animal FMT Consortium [19], including normal appetite and stool, absence of recent illness or medication use, and negative results for intestinal parasites.

Prior to treatment, gut microbiota composition was assessed using 16S rRNA amplicon sequencing. Dogs with AHDS showed significantly reduced microbial diversity compared to healthy controls, as measured by the Shannon Diversity Index (SDI) (*p* = 0.002) [30]. The SDI is a commonly used metric that reflects both species richness and evenness in microbial communities, with lower values indicating a less diverse and potentially dysbiotic microbiome.

Despite rapid and comparable clinical recovery in all AHDS dogs—regardless of treatment group—microbiome shifts were observed in those who received FMT. At discharge, FMT recipients showed significant changes in gut microbial structure, with 18 operational taxonomic units (OTUs) increased relative to sham-treated dogs. An additional four OTUs remained elevated at 30-day follow-up [30].

Metabolomic analyses further supported these microbial shifts. Dogs with AHDS had reduced levels of short-chain fatty acid (SCFA)-producing bacteria, which are important for gut health. Notably, *Clostridium hiranonis*, a microbe involved in the conversion of primary to secondary bile acids, was depleted in AHDS dogs but increased in FMT recipients following treatment [30].

Although the small sample size limits statistical power, the study highlighted two important findings: (1) a clear difference in microbiota diversity between healthy dogs and those with AHDS, and (2) the ability of FMT to positively shift the gut microbiome—even in the absence of a clear clinical advantage. These results, in the context of earlier studies linking AHDS to microbial imbalances and *Clostridium perfringens* overgrowth [31], support continued investigation into the role of FMT in gastrointestinal disorders in dogs.

### 2.5. Chronic Enteropathy (CE) in Cats and Dogs

CE in companion animals includes two primary forms: chronic inflammatory enteropathy (CIE) and small cell gastrointestinal lymphoma (SCGL). CIE can be further subclassified based on treatment response into immunosuppressant-responsive enteropathy (IRE) and food-responsive enteropathy (FRE) [32]. Dysbiosis—an imbalance in the intestinal microbiota—has been reported in a significant proportion of animals with CE, affecting approximately 64% of dogs and 76% of cats [33].

Several reports have examined the use of FMT in cats with CE. In one case report, a cat with a 16-month history of gastrointestinal symptoms achieved a three-month remission following a single enema of fresh feces [34].

In a more rigorous investigation, Karra et al. [35] conducted a prospective, randomized study involving 28 cats: 19 with CIE and 9 with SCGL. Eleven cats received a single enema FMT, while 17 served as controls. Clinical outcomes were assessed using the Feline Chronic Enteropathy Activity Index (FCEAI), and intestinal dysbiosis was evaluated using a feline dysbiosis index at baseline and 30 days post-treatment. Although FMT did not result in statistically significant differences in dysbiosis scores, clinical improvement was observed across all FMT-treated cats, with reductions in FCEAI scores. The authors acknowledged that additional interventions—such as dietary changes and adjunctive medications—may have influenced outcomes. They also suggested that sample size limitations and the use of a single FMT dose could have contributed to the lack of statistically significant microbiome changes. Despite these limitations, the study demonstrated that FMT was well tolerated, with no signs of clinical worsening, and offers preliminary support for its use in feline CE [35].

### 2.6. Safety Considerations for Companion Animal FMT

Although FMT shows promise in managing chronic enteropathies and other gastrointestinal conditions in companion animals, safety remains a critical concern. Until rigorously standardized, commercially available products become more widely accessible, strict donor screening is essential to minimize risk.

Donor feces should be screened for parasites transmissible via the fecal–oral route, including Giardia, Cryptosporidium, hookworms, whipworms, and roundworms. Testing should include both ova and adult forms. Other pathogens of concern include *Clostridium perfringens*, *Clostridioides difficile*, *Campylobacter coli*, *Campylobacter jejuni*, *Salmonella* spp., canine distemper virus, canine enteric coronavirus, canine parvovirus-2, and canine circovirus. Additionally, donors should be evaluated for recent antibiotic exposure, antimicrobial resistance, and recent hospitalization.

Comprehensive screening recommendations are outlined in the clinical practice guidelines developed by the Companion Animal FMT Consortium [19]. These guidelines offer evidence-based recommendations for FMT implementation in clinical veterinary settings and emphasize the increasing feasibility of FMT in companion animal practice. However, further controlled studies are needed to optimize dosing, delivery routes, and long-term outcomes.

## 3. FMT Use in Feed Animals

FMT is gaining attention in livestock (or “feed” animals) for its potential to improve animal health, reduce infection burden, and enhance productivity. Reported benefits include treatment of gastrointestinal infections, prevention of pathogen spread, and optimization of growth and feed efficiency.

In modern pig farming, widespread use of disinfection, high-pressure washing, and early-life animal isolation has unintentionally led to disrupted gut microbiomes and loss of native microbial diversity [36]. Rahman et al. [36] hypothesized that such farming practices cause a detrimental shift in microbial composition and evaluated whether microbiota from wild boars—more reflective of a natural microbial environment—could help restore balance in piglets.

Their study involved a control group and four treatment arms: (1) wild boar (WB) microbiota only, (2) mixed microbial cohorts (MMC), (3) WB plus sow-derived material, and (4) sow-derived microbiota only. Rahman et al. reported on four of the five groups. Data from the sow-derived microbiota was not reported. Piglets in the WB and MMC groups showed better microbial colonization by WB-derived bacteria. Cecal metabolite analysis at day 48 revealed that WB-treated piglets were enriched in histamine, acetyl-ornithine, ornithine, and citrulline—metabolites linked to metabolic and immune signaling. Elevated histamine levels were positively associated with Lactobacillus abundance [36].

The WB inoculum exhibited a numerically higher bacterial diversity and richness (based on observed and Shannon indices) than the Sow inoculum, although it had lower evenness based on Simpson indices. The Mix inoculum had higher Shannon diversity than the WB and Sow inocula and showed higher Simpson evenness than the WB inoculum. At the family level, the WB inoculum was enriched with *Peptostreptococcaceae*, *Lactobacillaceae*, *Tannerellaceae*, *Lachnospiraceae*, *Veillonellaceae*, and *Erysipelatoclostridiaceae*, whereas the Sow inoculum was enriched with *Clostridiaceae*, *Bacteroidaceae*, *Enterobacteriaceae*, *Streptococcaceae*, *Erysipelatoclostridiaceae*, and *Veillonellaceae*.

Linear discriminant analysis (LDA) effect size (LEfSe) package in R identified differentially abundant taxonomical features among treatment groups (*p* < 0.05). LEfSe revealed bacteria enrichment patterns on day 6 post-transplantation. WB inoculum recipients were enriched with a group of bacteria associated with fiber degradation and SCFA production, including *Prevotella*, *Treponema porcinum*, *Prevotellaceae*, *Selenomonas*, *WPS-2*, *Elusimicrobium*, *Bifidobacterium adolescentis*, *Bifidobacterium thermophilum*, *Veillonella caviae*, and *Selenomonadaceae*. Sow inoculum recipients also had increased levels of carbohydrate-fermenting bacteria, including *Frisingicoccus*, *Sutterella*, *Lachnospiraceae UCG-003*, *Eubacterium coprostanoligenes* group, uncultured *Ruminococcaceae*, *Solobacterium*, and *Mailhella*. Interestingly, control piglets displayed a higher abundance of potential pathogens such as Campylobacter, and carbohydrate-fermenting taxa such as *Desulfovibrio*, *Erysipelotrichaceae*, *Eubacterium hallii* group, and *Alloprevotella* (FDR *p* < 0.05). The Mix group showed similar enrichment patterns to the WB and Sow groups, with elevated levels of *Clostridia*, *Eubacterium xylanophilum* group, and RF39.

On day 28 post-transplantation (PND 48), the overall fecal microbial community structure was altered (Control: *n* = 12; Sow: *n* = 10; WB: *n* = 12; and Mix: *n* = 12), with differences detected by β-diversity analysis as shown in Figure 2 (Adonis, R^2^ = 0.08, *p* = 0.04; Betadispersion *p* = 0.50). On day 28 post-transplantation (PND 48), α-diversity analysis indicated that the mix group had lower species richness (Observed) and diversity (Shannon) than Control and WB groups (KW test, Observed *p* = 0.009; Shannon *p* = 0.04; Simpson *p* = 0.20). WB inoculum-derived unique ASVs were more successful in colonizing the piglet gut (Figure 3).

FMT also impacted immune markers. Piglets in the WB group had significantly lower cecal concentrations of IL-1β and IL-6 compared to both control and sow-only groups. The MMC group showed reduced IFN-γ, IL-2, and IL-6 relative to the sow group [36]. Shifts in microbial community structure were confirmed by beta-diversity analyses. The WB group showed significant divergence from controls (Adonis *p* = 0.02), and a trend toward divergence from the sow-only group (*p* = 0.07), but not from the MMC group (*p* = 0.23). These findings suggest that early-life FMT using wild boar-derived microbes can effectively reshape gut microbiota and immune profiles without adverse effects [36].

The authors showed that a successful colonization of wild boar-derived microbes to domestic pigs early in life via cultured MMC can alter gut microbial communities as well as cecal metabolite and cytokine profiles without causing adverse effects. Further research is needed to explore the relationship between these microbial-associated metabolites and immune modulation. Additionally, infection challenge studies are warranted to evaluate the disease resistance potential of these microbial communities. Although this study used 16S rRNA sequencing, the authors noted that additional -omics tools (e.g., metagenomics, metabolomics) could provide deeper insights into microbial function. Larger-scale studies are also needed to validate these findings and assess long-term health and productivity outcomes.

Chen et al. [37] conducted a broader review of FMT use in agricultural species and reported findings from 14 studies across chickens, pigs, cows, steers, and ruminants. Across these species, FMT was shown to affect growth rates, gut health, immune function, and in some cases, resistance to enteric infections (Figure 4). The authors emphasized that enhancing animal microbiota is not only beneficial for productivity but also for welfare, arguing that healthier, more resilient animals tend to be more efficient and require fewer interventions.

To frame animal welfare outcomes, Chen et al. referenced the framework developed by Fraser et al. [38], which outlines three guiding principles for ethically evaluating welfare interventions:1.Animals should be allowed to live natural lives, expressing their adaptations and capabilities.2.Animals should feel well, avoiding prolonged pain, fear, or distress and experiencing normal pleasures.3.Animals should function well, maintaining good health, growth, and behavioral and physiological stability.

These principles highlight that FMT’s value may extend beyond health and yield to include ethical and welfare dimensions—a consideration increasingly important to producers, veterinarians, and consumers alike. Importantly, these outcomes must be assessed with scientific rigor, as the link between microbial intervention and long-term health or welfare is complex. The balance of the outcomes of differing welfare is highlighted in Figure 4, where examples of both good and poor are shown. Some of these characteristics are difficult to assess in animals, others are clear such as health status, response to stress and other lesser-defined features. Moreover, there are host factors that can impair the ability of sick animals to respond to stress in an efficacious manner.

A different approach to microbiome reconstitution in dairy cows was investigated by Huang et al. [39], who employed both 16S rRNA-based metagenomics and serum non-targeted metabolomics to assess microbial and metabolic changes following FMT. In this study, 24 prepartum dairy cows were randomly assigned to one of three groups: saline infusion (control), fresh rumen fluid (FR), or stabilized rumen fluid (SR) administered after calving. Fresh rumen fluid (FR) contains live microbes, leading to higher gas production and volatile fatty acid (VFA) generation during fermentation, but its high microbial activity can limit incubation time. Stabilized rumen fluid (SR), on the other hand, undergoes sterilization to remove microbes and may have chemical buffers added to control pH, which reduces microbial activity and can alter fermentation patterns but also prevents disease transmission and allows for longer and more consistent in vitro incubations. While there were no significant changes in overall microbial richness across groups, shifts in the relative abundance of specific taxa were observed. These microbial changes were accompanied by alterations in metabolic profiles, particularly in amino acids, bile acids, and fatty acids.

Notably, both FR and SR groups showed increased abundance of *Clostridium butyricum*, a beneficial butyrate-producing species important for maintaining gut integrity. Simultaneously, there was a reduction in *Eubacterium ventriosum*, a bacterium with potential pathogenicity. These findings suggest that FMT with either fresh or stabilized rumen contents may enhance hindgut health and support liver metabolic function, likely due to improved volatile fatty acid (VFA) profiles and reduced pathogenic load [39].

## 4. FMT in Captivity or Conservation

The use of FMT in conservation medicine is an emerging approach to improve animal health and welfare in captive settings and potentially enhance reproductive success in endangered species. A review by Burnham et al. [40] examined the gut microbiomes of rhinoceros species, among the most endangered terrestrial animals globally. The family Rhinocerotidae comprises five extant species: the white rhinoceros (*Ceratotherium simum*), black rhinoceros (*Diceros bicornis*), greater one-horned or “Indian” rhinoceros (*Rhinoceros unicornis*), Sumatran rhinoceros (*Dicerorhinus sumatrensis*), and Javan rhinoceros (*Rhinoceros sondaicus*). These species face varying degrees of extinction risk, driven primarily by poaching for their horns and, secondarily, by habitat fragmentation that disrupts reproductive opportunities [41].

It is important to acknowledge the potential effect of captivity on baseline microbiota. A comprehensive study of 41 species ranging from canines, giraffe, anteaters, monkeys, lemurs and aardvarks [42]. Differences in bacterial richness were reported between wild and captive states. Captivity represents a significant shift in lifestyle for many animals. These data were collected from a range of sites where differences in feed and medical practices were evident. Some species exhibited a greater change such as primates while carnivores showed less of a shift.

A review by Dallas and Warne [43] examined the role of the microbiota, including bacteria, viruses, and fungi in captivity compared to wild populations. They discuss that many conditions or exposures in captivity, including antibiotic exposure, altered diet composition, homogeneous environment, and increased stress can alter the gut microbiome. It is clear that captive breeding programs are critical for population preservation but face challenges such as infertility, early embryonic loss, irregular estrous cycles, and diseases uncommon in wild populations (e.g., hemolytic anemia, renal failure) [44]. Gut microbiome differences between wild and captive rhinoceroses may underlie some of these health and reproductive issues. Roth et al. [45] conducted a comparative study of four rhinoceros species (*n* = 31 total) and found that, despite host species differences, the gut microbiome was generally dominated by *Firmicutes* (51–66.3%) and *Bacteroidetes* (23.4–39.8%), followed by *Verrucomicrobia* (1.9–7.6%), *Spirochetes* (1.1–3.1%), *Actinobacteria* (0.03–1.04%), and *Fibrobacteres* (0.19–2.14%).

However, microbiome research in wild animals faces substantial logistical obstacles as shown in Figure 5.

Burnham et al. [40] compiled methodological pitfalls from 2013 to 2022, including delays in fecal collection post-defecation, which can significantly alter microbial composition—especially in hot, humid climates. For example, samples stored at room temperature without preservation show measurable shifts in diversity and abundance within 24 h [40,46]. To improve reproducibility and utility in conservation programs, Burnham et al. advocate for standardized sample collection, microbial profiling (-omics), and the creation of a global microbiome reference database [40].

One illustrative case of microbiome manipulation in captivity is reported by Thacher et al. [47], who described FMT in a two-toed sloth housed in a U.S. Zoo. Sloths typically defecate every 7–10 days, making deviations from this pattern an early indicator of gastrointestinal distress. In this case, a healthy male sloth served as the donor for a female with altered fecal output. After three FMTs—two of which were accepted—the recipient’s stool normalized, and microbiome analysis revealed progressive shifts toward the donor’s profile. No adverse effects were reported, and normal fecal function was sustained for at least seven months.

A similar outcome was observed in koalas by Blyton et al. [48], who transferred microbiota from wild koalas (with a varied diet including *Eucalyptus obliqua*) into captive individuals accustomed only to manna gum. Recipients of the wild-derived microbiota exhibited shifts in gut composition and increased consumption of *E. obliqua*, suggesting that microbiota changes may drive adaptive dietary behavior.

Bornbusch and colleagues [49] reported that the impact of FMT after antibiotic therapy in cheetahs. A study examined 21 cheetahs using autologous FMTs examining the impact of FMT post-antibiotic recovery. It is known that antibiotics reduced that abundance of bacteria and FMT quickened post-antibiotic recovery. It was postulated that engraftment of bacteria facilitates production of beneficial metabolites.

Linnehan et al. [50] evaluated the safety and efficacy of FMT in bottlenose dolphins using metagenomic sequencing. Gastrointestinal diseases are a major cause of morbidity in bottlenose dolphins. Few studies have been conducted in this species, notably the utility of FMT in wild and managed dolphins. However, there are major challenges in collecting feces to study using these techniques such as non-firm feces, plus the actual collection of samples from healthy and sick animals. Fourteen dolphins six were healthy donors, six male and one female with an age range of 7 to 46 years. A slurry of feces from 7 donors was made and administration was inserted via the rectum via a catheter to a depth of 20 inches. Fecal samples were collected from each recipient 2–6 day post FMT and biweekly over a 6–13-week period. Each dolphin received eight FMTs over an eight-week period. Each dolphin feces was examined using metagenomic methods. Comparison of feces from healthy donors and sick dolphins FMT led to a positive shift in the sick metagenome to the normal healthy fecal profiles. Clearly administering FMT to aquatic creatures is challenging but the methods described provide a guide to the process.

Further studies of lemurs [51], lizards [52], caribou calves [53], and sifakas [54] have been published each demonstrating further perspectives on FMT and management of conservation.

Together, these studies highlight the potential of FMT as a non-invasive strategy to improve health, adaptability, and welfare in captive or endangered animals. However, further controlled trials, standardized protocols, and cross-species data sharing are needed to fully realize its utility in conservation biology.

## 5. Exploration of Human Conditions Using FMT in Animals

Animal models are increasingly recognized as valuable tools for investigating the human microbiome and its influence on health and disease. Recent reviews have highlighted that the integration of animal models with multi-omics technologies enables comprehensive study of complex processes, including host–microbe interactions, immune responses, and metabolic disorders. The effectiveness of these models depends heavily on the choice of animal species, experimental design, and the degree of physiological similarity and microbial compatibility with humans. Notably, diverse animal models—including rodents, pigs, and companion animals like dogs—offer distinct advantages for exploring the etiology and pathogenesis of inflammatory and metabolic diseases [55].

Over the past 10–15 years, research using animal models has significantly expanded our understanding of conditions such as obesity, inflammatory bowel disease, and neurodegenerative disorders. However, traditional models—particularly rodents—have limitations. Mice, for instance, differ markedly from humans in gut anatomy and physiology, and their ability to support colonization by human-derived microbes is limited.

To overcome these constraints, researchers have turned to non-traditional models. Examples include:Hydra, for studying microbial regulation of gut epithelial peristalsis [56];Squid, for insights into circadian rhythm regulation [57];Aphids, to investigate metabolite exchange [58,59];Killifish, which have been used to explore microbial influences on lifespan [60].

Non-human primates offer a particularly promising model due to their genetic and physiological similarities to humans. The common marmoset (*Callithrix jacchus*), for example, has become an important model for studying aging, with a lifespan (~10–15 years) that bridges the gap between rodents and longer-lived primates, like chimpanzees. Compared to other primates, marmosets present several age-related pathologies that closely mirror those in humans, making them well-suited for exploring the microbiome’s role in health span and disease progression [61,62].

Reveles et al. [63] conducted a metagenomic analysis comparing the gut microbiomes of young adult and geriatric marmosets. Using 16S rRNA V4 region sequencing, they analyzed microbial diversity through operational taxonomic units (OTUs) and the Shannon Diversity Index. Results showed that geriatric marmosets had significantly lower microbiome diversity than younger counterparts (3.15 vs. 3.46; *p* = 0.0191), with a higher mean abundance of Proteobacteria (0.22 vs. 0.09; *p* = 0.0233) and a lower abundance of Firmicutes (0.15 vs. 0.19; *p* = 0.0032). Notably, three older animals exhibited microbial profiles more closely aligned with the younger group, highlighting inter-individual variability.

Ross et al. [64] further demonstrated the feasibility of FMT in this model. In their study, healthy young adult marmosets received weekly oral FMT via unsedated gavage over three weeks. The procedure was well tolerated, with no adverse effects observed during a six-month follow-up. Microbiome sequencing confirmed durable shifts in microbial composition, supporting the use of FMT in marmosets as a tool to investigate microbiome-mediated aspects of aging.

Together, these findings underscore the value of diverse animal models—including primates—for mechanistic studies of the human gut microbiome and its potential therapeutic modulation via FMT.

## 6. Conclusions

Advancements in our understanding of the microbiome—particularly within the gastrointestinal tract—are driving meaningful improvements in the health management of animals across a broad spectrum of species and settings. As analytical techniques continue to evolve, the range of conditions in which FMT and microbiome modulation may be applied is expanding rapidly.

In human medicine, it is well established that the gut microbiome plays a central role in systemic health, influencing not only gastrointestinal function but also the brain, liver, and other organs through pathways such as the gut–brain and gut–liver axes. These complex interactions are increasingly being recognized in veterinary and zoological medicine as well, with implications for companion animals, livestock, and wildlife in managed care or rehabilitation programs.

Furthermore, animal models—particularly non-human primates like marmosets—are proving invaluable in exploring microbiome-related aspects of human health, including aging and chronic disease. These parallels support the growing relevance of FMT in translational research.

Despite the establishment of dysbiosis morbidity is not always a consequence and clearly there are many challenges to developing an FMT procedure for collecting, screening and administration of the products. In humans we are beginning to appreciate the complex role of the gastrointestinal microbiome and the role dysbiosis has in several non-gastrointestinal conditions such as depression, liver diseases and a multitude of other emerging diseases. To the authors’ knowledge there have been no studies evaluating non-gastrointestinal diseases in animals.

This review has highlighted the emerging applications of FMT in veterinary medicine and conservation biology. The evidence demonstrates both the promise and complexity of applying microbiome-based interventions across species, contributing not only to animal health and welfare but also to broader insights that may benefit human medicine. These publications present a glimpse into the potential use and outcomes of FMT in a range of animals in different settings, however there is a need for larger scale studies in several cases.

## Figures and Tables

**Figure 1 microorganisms-13-02465-f001:**
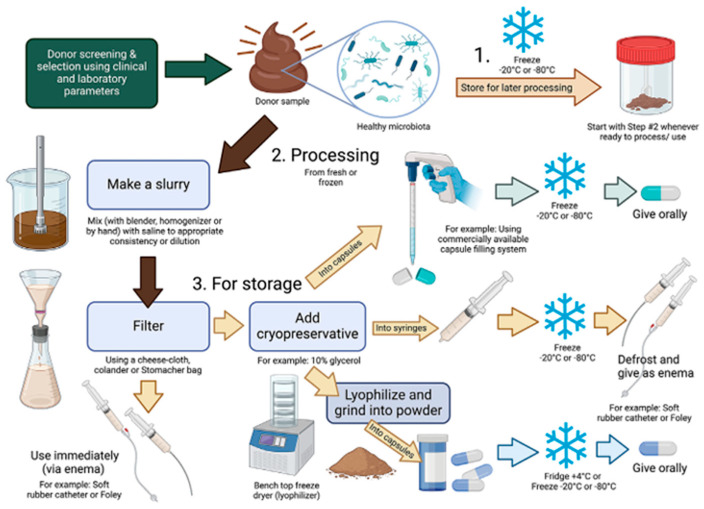
Overview of FMT product preparation and processing (reproduced from [19]). After a fecal donor screening and selection is complete, naturally voided feces can be used for FMT preparation and processing. Freshly void feces is ideal for FMT administration. If use of fresh feces is not feasible, then feces can be stored until processing and/or use (Step 1). Processing steps for fresh feces and/or frozen feces including making a fecal slurry and filtering the FMT product (Step 2). Once the FMT product is prepared, it can be used immediately or can be stored (Step 3). If the FMT product is to be stored, a cryopreservative can be added. FMT capsules can be administered orally and processed fecal slurries can be administered via rectal enema (Created with BioRender.com).

**Figure 2 microorganisms-13-02465-f002:**
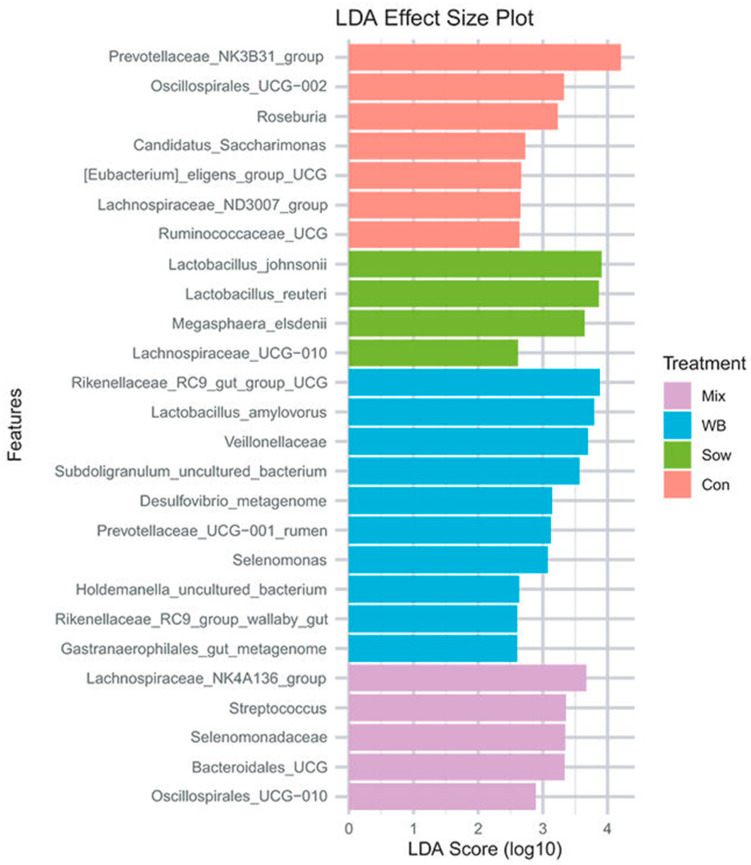
Initial piglet gut microbiota 28 days after FMT (reproduced from [36]). LDA, linear discriminant analysis; WB, wild boar; Con, control; Mix, mixed microbial cohort; Sow, sow-derived microbiota.

**Figure 3 microorganisms-13-02465-f003:**
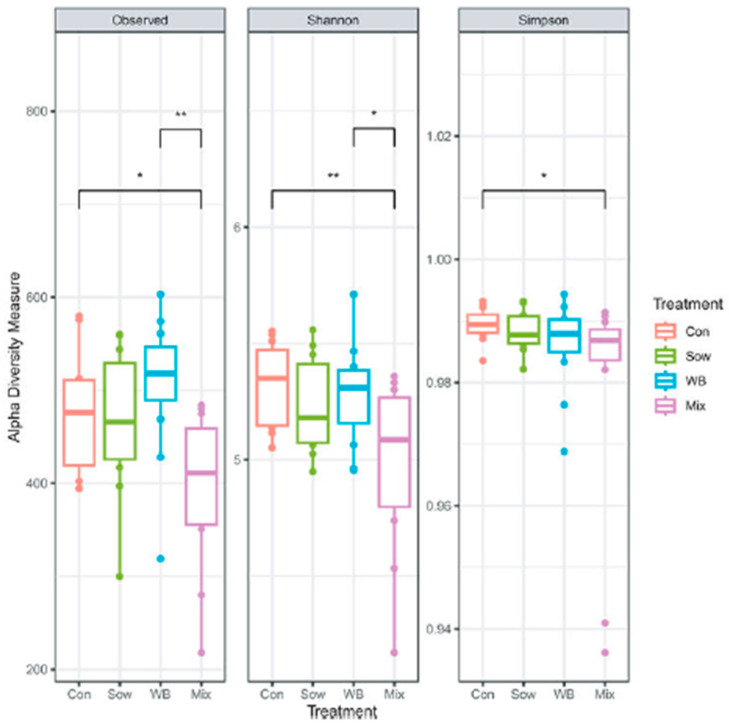
Changes over 28 days of the microbiota of the four groups (reproduced from [36]). * *p* < 0.05; ** *p* < 0.01.

**Figure 4 microorganisms-13-02465-f004:**
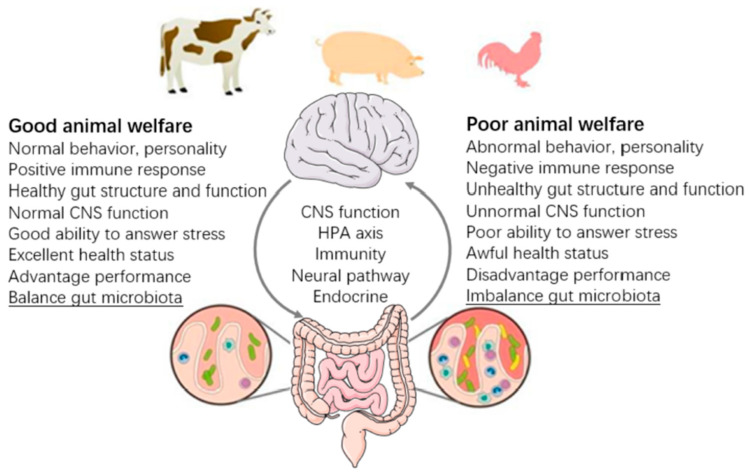
Animal welfare in the implication and perspective of the gut microbiome (reproduced with permission from [37]).

**Figure 5 microorganisms-13-02465-f005:**
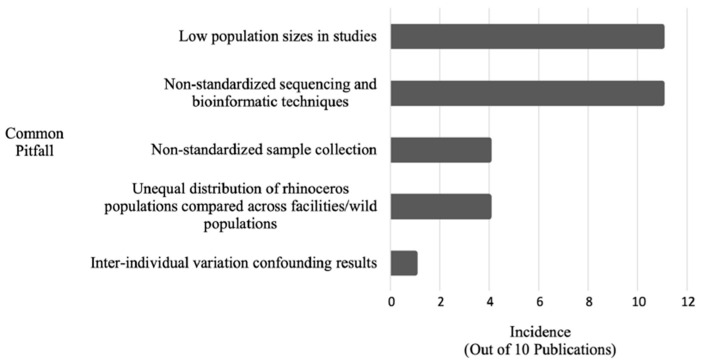
Incidence of common pitfalls within *n* = 11 rhinoceros gut microbiome publications published between 2013 and 2022 (reproduced from [40]).

## Data Availability

No new data were created or analyzed in this study. Data sharing is not applicable to this article.

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
