# Peer review of "Fecal Microbiota Transplantation in Animals: Therapeutics, Conservation, and Farming"

_microorganisms, 2025, doi:10.3390/microorganisms13112465_

Round 1
Reviewer 1 Report
Comments and Suggestions for Authors
This review discusses the use of fecal microbiota transplants (FMTs) in animal medicine. The topic is timely and valuable. The review is well written, and I only have a few comments.
Figure 1. It is important to note that this is the recommended ‘clinical’ protocol for FMTs, but not all FMTs are prepared/administered this way. I think it is valuable to note that the preparation and administration of FMTs can vary widely across species, particularly in non-domesticated hosts. This figure provides some depictions of this but it is not really discussed in the text. In other words, while feces is readily available, the equipment for clinical preparation may be limited at some facilities (e.g., zoos). Similarly, donor selection for exotic species can be more challenging than for humans/domesticated animals. So, is often a tradeoff between optimization and accessibility that is worth highlighting.
Figure 2. This figure is a little confusing. If I understand correctly, there are 5 treatment groups: (1) control, (1) wild boar (WB) microbiota only, (2) mixed microbial cohorts (MMC), (3) WB plus sow-derived material, and (4) sow-derived microbiota only. This LDA plot seems to be showing the log-fold change scores for four of the groups, each seemingly compared to the WB plus sow-derived material - that is the group missing from the plot and so I’m assuming it is the ‘baseline’ group to which all other groups are compared? But wouldn’t you want to report the four treatment groups compared to the control group? I am not sure this figure adds much to the paper.
FMT in captivity or conservation. I have a couple comments about this section, namely that there are significantly more studies that can add to this section.
- Lines 313-342: Why the focus on rhinos? This seems very specific and out of place. I appreciated the need to establish that captivity is a substantial driver of microbiota variation, but there are numerous reviews that more broadly discuss this trend across many host species. It would better serve this review to include more broad discussion of these issues, rather than focusing on one group of hosts. See these examples:
- Dallas, J. W., & Warne, R. W. (2023). Captivity and animal microbiomes: potential roles of microbiota for influencing animal conservation. Microbial ecology, 85(3), 820-838.
- Bornbusch, S. L., Power, M. L., Schulkin, J., Drea, C. M., Maslanka, M. T., & Muletz‐Wolz, C. R. (2024). Integrating microbiome science and evolutionary medicine into animal health and conservation. Biological Reviews, 99(2), 458-477
- McKenzie, V. J., Song, S. J., Delsuc, F., Prest, T. L., Oliverio, A. M., Korpita, T. M., ... & Knight, R. (2017). The effects of captivity on the mammalian gut microbiome. Integrative and comparative biology, 57(4), 690-704
- Although the literature on FMTs in ‘wildlife’ or ‘exotic’ species is certainly less robust than the literature on humans and domesticated species, there are significantly more studies (included case studies) that can bolster this section of the review. I suggest the authors do an updated search but here are a few to start:
- Linnehan BK, Kodera SM, Allard SM, Brodie EC, Allaband C, Knight R, Lutz HL, Carroll MC, Meegan JM, Jensen ED. 2024. Evaluation of the safety and efficacy of fecal microbiota transplantations in bottlenose dolphins (Tursiops truncatus) using metagenomic sequencing. J Appl Microbiol 135:lxae026.
- Bornbusch, S. L., Crosier, A., Gentry, L., Delaski, K. M., Maslanka, M., & Muletz-Wolz, C. R. (2024). Fecal microbiota transplants facilitate post-antibiotic recovery of gut microbiota in cheetahs (Acinonyx jubatus). Communications Biology, 7(1), 1689.
- Kohl KD, Weiss RB, Cox J, Dale C, Denise Dearing M. 2014. Gut microbes of mammalian herbivores facilitate intake of plant toxins. Ecol Lett 17:1238–1246.
- Yang J, Liu W, Han X, Hao X, Yao Q, Du W. 2024. Gut microbiota modulation enhances the immune capacity of lizards under climate warming. Microbiome 12:37
- Bornbusch SL, Harris RL, Grebe NM, Roche K, Dimac-Stohl K, Drea CM. 2021. Antibiotics and fecal transfaunation differentially affect microbiota recovery, associations, and antibiotic resistance in lemur guts. Anim Microbiome 3:65.
- McKenney, E. A., Greene, L. K., Drea, C. M., & Yoder, A. D. (2017). Down for the count: Cryptosporidium infection depletes the gut microbiome in Coquerel’s sifakas. Microbial Ecology in Health and Disease, 28(1), 1335165.
- Denryter, K., & Beckmen, K. B. (2024). Suspected Case of Persistent Thiamin Deficiency in a Hand‐Reared Caribou Calf. Zoo Biology, 43(6), 580-584
- Stapleton TE, Kohl KD, Dearing MD. 2022. Plant secondary compound-and antibiotic-induced community disturbances improve the establishment of foreign gut microbiota. FEMS Microbiol Ecol 98:fiac005.
Author Response
Comments 1: Figure 1. It is important to note that this is the recommended ‘clinical’ protocol for FMTs, but not all FMTs are prepared/administered this way. I think it is valuable to note that the preparation and administration of FMTs can vary widely across species, particularly in non-domesticated hosts. This figure provides some depictions of this but it is not really discussed in the text. In other words, while feces is readily available, the equipment for clinical preparation may be limited at some facilities (e.g., zoos). Similarly, donor selection for exotic species can be more challenging than for humans/domesticated animals. So, is often a tradeoff between optimization and accessibility that is worth highlighting.
Response 1: Thank you for this insightful comment. The text has been modified to accommodate account for these challenges.
Comments 1: Figure 2. This figure is a little confusing. If I understand correctly, there are 5 treatment groups: (1) control, (1) wild boar (WB) microbiota only, (2) mixed microbial cohorts (MMC), (3) WB plus sow-derived material, and (4) sow-derived microbiota only. This LDA plot seems to be showing the log-fold change scores for four of the groups, each seemingly compared to the WB plus sow-derived material - that is the group missing from the plot and so I’m assuming it is the ‘baseline’ group to which all other groups are compared? But wouldn’t you want to report the four treatment groups compared to the control group? I am not sure this figure adds much to the paper.
Response 1: Thank you for identifying this weakness. We have expanded this section to provide clarity on this pertinent study.
Comments 1: FMT in captivity or conservation. I have a couple comments about this section, namely that there are significantly more studies that can add to this section.
- Lines 313-342: Why the focus on rhinos? This seems very specific and out of place. I appreciated the need to establish that captivity is a substantial driver of microbiota variation, but there are numerous reviews that more broadly discuss this trend across many host species. It would better serve this review to include more broad discussion of these issues, rather than focusing on one group of hosts. See these examples:
- Dallas, J. W., & Warne, R. W. (2023). Captivity and animal microbiomes: potential roles of microbiota for influencing animal conservation. Microbial ecology, 85(3), 820-838.
- Bornbusch, S. L., Power, M. L., Schulkin, J., Drea, C. M., Maslanka, M. T., & Muletz‐Wolz, C. R. (2024). Integrating microbiome science and evolutionary medicine into animal health and conservation. Biological Reviews, 99(2), 458-477
- McKenzie, V. J., Song, S. J., Delsuc, F., Prest, T. L., Oliverio, A. M., Korpita, T. M., ... & Knight, R. (2017). The effects of captivity on the mammalian gut microbiome. Integrative and comparative biology, 57(4), 690-704
- Although the literature on FMTs in ‘wildlife’ or ‘exotic’ species is certainly less robust than the literature on humans and domesticated species, there are significantly more studies (included case studies) that can bolster this section of the review. I suggest the authors do an updated search but here are a few to start:
- Linnehan BK, Kodera SM, Allard SM, Brodie EC, Allaband C, Knight R, Lutz HL, Carroll MC, Meegan JM, Jensen ED. 2024. Evaluation of the safety and efficacy of fecal microbiota transplantations in bottlenose dolphins (Tursiops truncatus) using metagenomic sequencing. J Appl Microbiol 135:lxae026.
- Bornbusch, S. L., Crosier, A., Gentry, L., Delaski, K. M., Maslanka, M., & Muletz-Wolz, C. R. (2024). Fecal microbiota transplants facilitate post-antibiotic recovery of gut microbiota in cheetahs (Acinonyx jubatus). Communications Biology, 7(1), 1689.
- Kohl KD, Weiss RB, Cox J, Dale C, Denise Dearing M. 2014. Gut microbes of mammalian herbivores facilitate intake of plant toxins. Ecol Lett 17:1238–1246.
- Yang J, Liu W, Han X, Hao X, Yao Q, Du W. 2024. Gut microbiota modulation enhances the immune capacity of lizards under climate warming. Microbiome 12:37
- Bornbusch SL, Harris RL, Grebe NM, Roche K, Dimac-Stohl K, Drea CM. 2021. Antibiotics and fecal transfaunation differentially affect microbiota recovery, associations, and antibiotic resistance in lemur guts. Anim Microbiome 3:65.
- McKenney, E. A., Greene, L. K., Drea, C. M., & Yoder, A. D. (2017). Down for the count: Cryptosporidium infection depletes the gut microbiome in Coquerel’s sifakas. Microbial Ecology in Health and Disease, 28(1), 1335165.
- Denryter, K., & Beckmen, K. B. (2024). Suspected Case of Persistent Thiamin Deficiency in a Hand‐Reared Caribou Calf. Zoo Biology, 43(6), 580-584
- Stapleton TE, Kohl KD, Dearing MD. 2022. Plant secondary compound-and antibiotic-induced community disturbances improve the establishment of foreign gut microbiota. FEMS Microbiol Ecol 98:fiac005.
Response 1: These references are very helpful. Thank you. We have incorporated a number of these studies and their corresponding data to provide further insights in the use of FMT in captive animals.
Reviewer 2 Report
Comments and Suggestions for Authors
The manuscript entitled “Fecal microbiota transplantation in animals: therapeutics, conservation, and farming” is a well-written and informative paper. The Authors presented the role of FMT in humans in a comprehensive way, indicating its value in restoring the gut microbiota. However, the main topic of the manuscript is the implementation of FMT in veterinary medicine. The manuscript's strong point is the indication of FMT as a potential therapy to restore gut composition and, consequently, control animal health. Moreover, the Authors focused not only on companion animals but also on the FMT treatment of livestock and animals in captivity or conservation. However, it is essential to note that FMT transfer is not a cure for every disease. Moreover, the Authors should expand the fragment regarding the safety of FMT therapy in animals, even though they indicated the reference for the recommendations. But the review is informative, and it should provide more information regarding the safety of the treatment. This is a minor remark. I recommend the manuscript for publication.
Author Response
Comments 2: The manuscript entitled “Fecal microbiota transplantation in animals: therapeutics, conservation, and farming” is a well-written and informative paper. The Authors presented the role of FMT in humans in a comprehensive way, indicating its value in restoring the gut microbiota. However, the main topic of the manuscript is the implementation of FMT in veterinary medicine. The manuscript's strong point is the indication of FMT as a potential therapy to restore gut composition and, consequently, control animal health. Moreover, the Authors focused not only on companion animals but also on the FMT treatment of livestock and animals in captivity or conservation. However, it is essential to note that FMT transfer is not a cure for every disease. Moreover, the Authors should expand the fragment regarding the safety of FMT therapy in animals, even though they indicated the reference for the recommendations. But the review is informative, and it should provide more information regarding the safety of the treatment. This is a minor remark. I recommend the manuscript for publication.
Response 2: Thank you for your kind comments. We have raised this point in the conclusion section.
Reviewer 3 Report
Comments and Suggestions for Authors
Major comments:
The current manuscript is a quite comprehensive and thorough review on the use of fecal microbiota transplantation.
Nevertheless, I miss a somewhat critical view of the procedure, weighthing its pros and cons.
The authors have discussed dysbiosis in some of the reviewed animal diseases. What about the others? Does not dysbiosis cause disease in every situation? Independent of its severity?
I would also expect the authors to discuss the gut as the second brain and how the gut microbiota diversity may impact animal health.
Minor comments:
Some of the images/figures are blurred or seem to have low resolution.
Author Response
Comments 3: Major comments: The current manuscript is a quite comprehensive and thorough review on the use of fecal microbiota transplantation. Nevertheless, I miss a somewhat critical view of the procedure, weighing its pros and cons.
Response 3: Thank you for your excellent suggestion. We have included text on the benefits and risks of the procedure.
Comments 3: The authors have discussed dysbiosis in some of the reviewed animal diseases. What about the others? Does not dysbiosis cause disease in every situation? Independent of its severity?
Response 3: Thank you for bringing this to our attention. We have included text which encompasses these points.
Comments 3: I would also expect the authors to discuss the gut as the second brain and how the gut microbiota diversity may impact animal health.
Response 3: We agree with your proposal; however, investigating some of the non-gastrointestinal effects could be quite difficult and beyond the scope of the current manuscript. We have included text to acknowledge this concept.
Comments 3: Minor comments: Some of the images/figures are blurred or seem to have low resolution.
Response 3: We have tried to gather a better resolution image where possible. I hope the journal can advise us of any problems.
Reviewer 4 Report
Comments and Suggestions for Authors
This is a well-prepared review on a critically relevant topic of gastrointestinal health and host wellness. The interplay of veterinary medicine with human medicine is well defined. The authors have effectively outlined the potential value of this work with appropriate caveats described to communicate the early stage of this area of study. As a review, this report provides the reader with properly referenced overview of the current status of this area of study. There is a need for larger scale investigations to be completed, the reported work provides a beneficial backdrop on what has been done in a variety of species to this point.
Line 89, should be listed as Figure 1 (vs. 1.3).
Line 264, Figure 2 will benefit from more comprehensive explanation that will improve clarity of this reported data.
Line 292, Figure 3. The caption for Poor animal welfare will benefit from revision.
Poor ability to answer stress, this is not clear and requires revision. Host factors impair ability to respond to stress in an effective manner would be more appropriate.
Awful health status is not appropriate. Poor health status would be more appropriate.
Line 298, the authors should define fresh rumen fluid and stabilized rumen fluid.
Author Response
Comments 4: This is a well-prepared review on a critically relevant topic of gastrointestinal health and host wellness. The interplay of veterinary medicine with human medicine is well defined. The authors have effectively outlined the potential value of this work with appropriate caveats described to communicate the early stage of this area of study. As a review, this report provides the reader with properly referenced overview of the current status of this area of study. There is a need for larger scale investigations to be completed, the reported work provides a beneficial backdrop on what has been done in a variety of species to this point.
Response 4: Thank you for this observation. We have stated the need for larger scale studies to improve our understanding of this potential beneficial management approach.
Comments 4: Line 89, should be listed as Figure 1 (vs. 1.3).
Response 4: Edited.
Comments 4: Line 264, Figure 2 will benefit from more comprehensive explanation that will improve clarity of this reported data.
Response 4: We have incorporated more data from the cited paper. We hope this answers your comment.
Comments 4: Line 292, Figure 3. The caption for Poor animal welfare will benefit from revision.
Response 4: We have modified this phrase.
Comments 4: Poor ability to answer stress, this is not clear and requires revision. Host factors impair ability to respond to stress in an effective manner would be more appropriate.
Response 4: We agree with your comment, so we have expanded this section to highlight this issue.
Comments 4: Awful health status is not appropriate. Poor health status would be more appropriate.
Response 4: Thank you, edited.
Comments 4: Line 298, the authors should define fresh rumen fluid and stabilized rumen fluid.
Response 4: Thanks for raising this topic. We have added an explanation of the difference between the 2 products.